# Challenges Facing the Delivery City Phenomenon after the COVID-19 Pandemic

**Li Won Kim**

The Graduate School of Environmental Studies, Seoul National University, Seoul 08826, Korea; gddfhfg@snu.ac.kr

**Abstract:** Due to the "untact" society caused by the COVID-19 pandemic, urban functions have grown increasingly reliant on the mobility of things rather than human mobility, giving rise to the "delivery city phenomenon" with real significance. This study explores the necessity to comprehend the demands and social views of delivery city inhabitants ("actors") to future urban planning, recognizing that the phenomenon of the delivery city has a significant impact on the shape and structure of modern cities. To do this, a grounded theory-based approach was used to evaluate keywords in Korean media articles and thoroughly code the content of actors' contextual interviews and shadowing data. The results indicate that the delivery city phenomenon has changed the geographical sense of actors and the role of each space, and has urban planning implications such as in the establishment of social and spatial infrastructure, the institutional basis for technological development, and the call for sustainability. This study is meaningful in understanding modern cities centered around delivery services, which have gained global prominence, and it can contribute to sustainable urban planning by deriving urban tasks for this new phenomenon.

**Keywords:** COVID-19; delivery city; mobility of things; urban planning; sustainability

## 1. Introduction

The market for delivery services, which was estimated at 36 billion units in 2013, increased 2.86 times to 103 billion units in 2019, equating to 3248 packages delivered every second. The annual delivery volume per person has almost tripled from 10 in 2013 to 34 in 2020 and is projected to double by 2026 with the advent of e-commerce platforms [1]. Such a substantial increase in volume indicates the dependence of urban functions on these services for the mobility of things across the world.

The delivery market size in Korea, which was the research target, exceeded KRW 15 trillion in 2018. As of 2021, there were 423,000 deliverers in Korea [2], and online shopping accounted for 62% as of January 2021 compared to offline shopping with purchases made directly from a [3]. Accordingly, it can be said that modern consumption behavior consists of purchases from online stores and delivery using "the mobility of things". The rapid growth of logistics since 2019 has been largely influenced by the non-face-to-face society triggered by COVID-19. Consumers are not required to visit a store physically to buy products, and consumption occurs in a non-face-to-face manner through online platforms that can be accessed at any time and from any location as long as a network connection is present. It was found that 74.7% of Seoul citizens have experienced non-face-to-face consumption activities after COVID-19. Additionally, 80% of consumers with non-face-to-face purchase experiences said that they would continue this lifestyle even after COVID-19 ended. Therefore, it can be seen that the function of Korean cities depends on the mobility of things. This study aims to understand the characteristics of modern cities through the framework of delivery service and thereby derive urban planning tasks. In this context, the term "delivery city" is used, which is the coined term for a "city" affected by the phenomenon of the mobility of things or "delivery" [4].

The work in [5] argued that "new risks" have emerged as a result of modernity. As transnational industries arise as a result of the process of urbanization, risks also move

along with its trajectory. In particular, COVID-19 brought to light the fact that risks are also shared globally, in the era of globalization and hyper-connection. With the advancement of technology, the trajectory of human mobility that transcends the physical limits of spaces has become a pathway of the virus, which has spread all over the world along with globalized mobility.

As human mobility has become an acute threat to human existence, it has been controlled and supervised globally to prevent its spread. As a consequence of the universalization of risks due to globalization, human mobility has been restricted in at-risk societies. As the pandemic limited human mobility, the mobility of things began to emerge in order to sustain the functionality of the city while reducing individual travel. Logistics services transport "things" such as food and daily necessities, which were formerly purchased directly by individual customers in the store, to the boundary of consumers' private spaces. The changes following COVID-19, such as the volume of logistics increasing by 36% and the average of 255 items delivered daily per deliveryman, demonstrate the pandemic-induced phenomenon of the delivery city.

This revolution of the mobility of things is spearheading the spatial geography and social transformation of the city by redefining the stakeholder relationships around the current commercial space. The author of [6] explored the idea of platform urbanism to examine how online services, such as Uber and Deliveroo, interact with locations and alter topography. The platform creates a new digital topography in the setting of the urban area that is connected with and influenced by the physical geography of the space. According to [7], the emergence of the delivery platform market calls for a need to investigate the changes in social arrangements that it has brought about, beyond the job of delivery workers. [7] also expresses that the platform functions as the "space for the articulation of different networks" where customers, restaurants, and riders could form "goods" like "delivered meals". The platform plays a role in market creation and should be understood from a broader social and economic perspective. The researcher of [8] also investigated the phenomena wherein a food delivery platform employee, a crucial player in a delivery city, uses place-making to change urban area into the geography of the workplace. It changes the geographical conditions of urban public spaces by delivery people.

However, recent urban studies surrounding delivery services have mainly focused on the labor conditions of delivery people and the spatial structure in which delivery platforms change. Therefore, this study aims to expand the subject of urban research according to the delivery city to urban residents, identify urban residents' needs and social perceptions of delivery services, and derive tasks for future urban planning. Research on the phenomenon of the delivery city and the needs of urban residents according to the continuously soaring demand for delivery services has timely appropriateness and justification.

In this research, in order to understand the phenomenon of the delivery city that supports the functions of modern society, the theoretical background and its flow were first reviewed, and then the social perception of a delivery city and the needs of urban residents were identified through a grounded theory approach. Following this, contextual interviews and spatial shadowing were carried out with 30 actors related to delivery services in order to qualitatively code their perspectives on the mobility of things. In addition, considering that the delivery city phenomenon is a relatively new urban phenomenon, this research attempted to examine the social interactions that this phenomenon produces. To perform this, a keyword analysis of 31,830 articles from 19 regional and economic daily newspapers in Korea was carried out. This study classified the research data into five themes using this framework for data analysis and, based on this classification, the study intended to determine the urban planning implications and tasks of a modern city where the mobility of things is widespread.

## 2. Literature Review

"Things" interact with humans [9–11]. Things are capable of human-like activity rather than existing as mere objects. In the course of their movement between places, they

change and reorganize places [9,12,13]. The relationship between things and spaces, as well as between objects and humans, changes places and spaces. As can be seen from the argument in [13] that the "sociology of things" has advanced over the last several years, things are becoming the primary subject of mobility study. The work in [13] classifies things with various mobility patterns according to differences in the relationship between things and the place of their origin as well as between things and the place of their destination. The majority of things in the phenomenon of the delivery city exist for mobility, despite the fact that their relationships with their origins and destinations are diverse. According to the work in [13], this category includes things that exist "in between". The origin and destination do not define the meaning of this category of things; rather, the process of mobility itself creates the meaning [13] (p. 79).

The approach for comprehending a city via the notion of mobility is closely related to the contrasting concepts of settlement and nomadism [12,14,15]. The work in [16] described the transformations in modern society that followed the advent of mobility as a task that even requires the redefinition of society. A local community, which seems to belong to a particular geography, is also the outcome of the reliance on diverse mobility. This mobility serves as a mechanism for the production and reproduction of social life and cultural forms, as well as a driving factor in the human occupation of cities [16]. As a consequence of the collaboration between transportation, science and technology, and urban planning, the concept of human mobility has been broadened, resulting in the realm of human mobility reaching the transnational level. Not only does the extension of human mobility improve the mobility of the human body, but it also increases the mobility of things that humans desire. Such mobility of things is generalized in modern society through shipping and delivery, representing a synthesis of the acceleration of capitalist market competitiveness, commodification of time and space, and flexibility of labor [17].

In a technologically evolved modern society, the mobility of things can be characterized as a modern nomadism as the mobility of things and the virtual mobility of information are maximized [18]. The authors of [18] stated that the modern society departs from the dualistic perspective of settlement and widens the meaning of nomadism by constructing a new system in which various heterogeneities exist and are interconnected [18] (p. 181). Based on the growth of network technology, modern cities are characterized by nomadic lifestyles [12,15,18], according to this extension of meaning. In such nomadic cities, the border between each space is not strictly delineated into space and place, but rather becomes liquid within the network [15,19].

In a contemporary city with mobile and nomadic characteristics, things, money, images, and people flow instantaneously and traverse boundaries via the technology of a hyperconnected society [14,20]. In addition, individuals in contemporary cities experience various modes of mobility, such as moving the body physically, moving things, or experiencing nomadism with a fixed body via technological advancements [12,14]. Various forms of mobility that can be experienced in modern cities coexist in urban space and construct a simultaneous movement system [21].

Even though the mobility of things is transforming space and place as a significant urban and social phenomenon, this topic has received only peripheral attention, such as research on logistics trends, spatial implications, and labor issues. In this context, the purpose of this research was to examine the delivery city phenomenon in three dimensions by integrating perspectives of space, actors, and society, and to determine the implications for urban planning.

## 3. Methodology

This study uses grounded theory [22,23] to understand the characteristics of modern cities where the conceptual framework of the delivery city phenomenon is as yet not clearly defined. First, the qualitative research methods of shadowing and contextual interview were conducted to obtain data on the needs of urban residents from a microscopic and practical perspective. The epistemological turn introduced into the mobility paradigm

resulted in methodological advancements in research. Since the concept of mobility includes movement, a new method of observation and research was required [12,24]. In this regard, the so-called "mobile" research method, which entails moving and observing with a "moving" study subject "together", has attracted attention [12,24–27]. Considering that the subject of mobility study changes, morphs, and moves [24], [28] (p. 10), the research approach also should be participatory and ethnographic, capturing the scene of fluid movement [19]. In this sense, this study was also conducted based on moving together with the research subject through co-present immersion, employing various observation and recording techniques, and using participatory interviews (Figure 1).

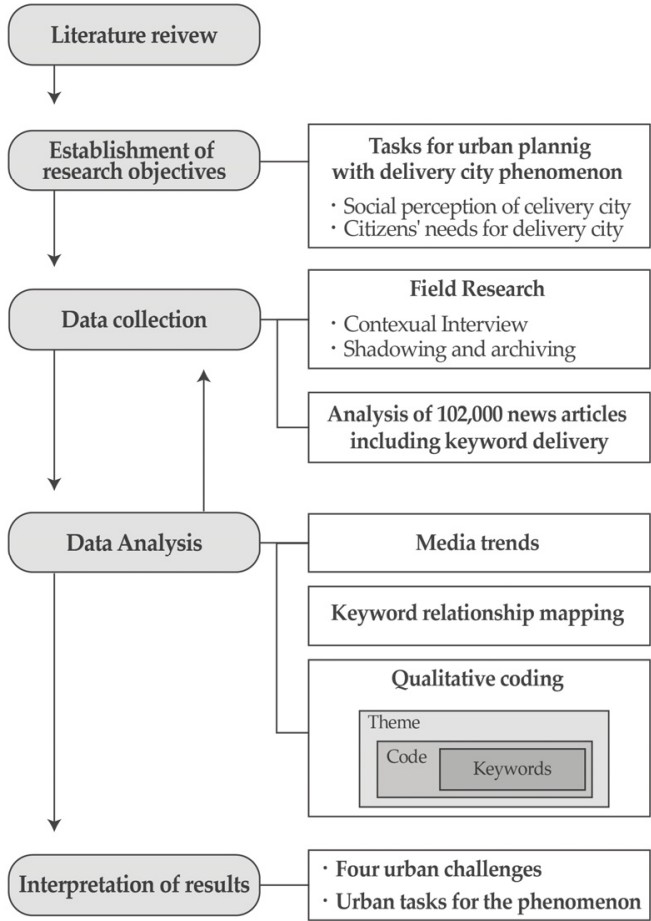

**Figure 1.** The grounded theory research process (source: created by the author).

The geographical scope of the research, as indicated in Figure 2, was the metropolitan region in Korea with the highest usage of delivery services. According to the categorization of the actors in the process of the mobility of things, the total number of participants in the interview was 29, which included 2 platform workers (Code-A), 5 warehouse workers (Code-B), 8 deliverers (Code-C), 10 receivers (Code-D), and 4 pedestrians (Code-E). This study uses contextual interviews, one of the in-depth interview methodologies. This is not simply interviewing actors but identifying complex and psychological factors rather than observing from an external point of view in the situation where the action occurs [29]. According to this methodology, questions were adjusted according to the interviewee's response after the first question, 'Please describe the most recent delivery experience' (Table 1). Interviews were conducted for about 30 min for each interviewee. Since the interview was conducted with Zoom in the COVID-19 situation, the customer interviewer was asked to open a delivery platform application to conduct a contextual interview in Zoom to help the context and immersion of the interview.

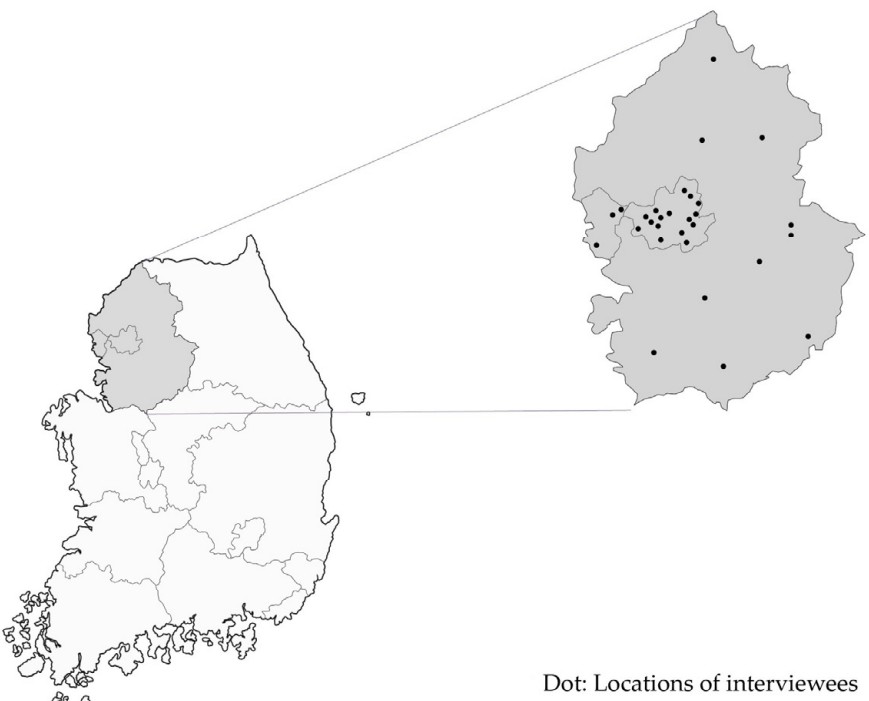

Dot: Locations of interviewees

**Figure 2.** The spatial scope of the study (source: created by the author).

**Table 1.** Topics of interview.

| Topics of Semi-Structured Interview |
| --- |
| 1. Could you please describe the most recent delivery experience? |
| 2. Could you explain your usual pattern of delivery use? include Location, frequency, etc. |
| 3. What is the key factor affects your usual pattern of delivery use? |
| 4. What do you think should be improved for better experience of delivery service? |

Data derived from qualitative studies were refined in two ways. In the case of interviews, each interview's data were backed up and then coded. Each of the derived codes was refined and connoted by stages such as raw data, code, and theme. Through the use of keywords commonly obtained from each participant's interview, this study attempted to investigate and interpret which topics that are related to the delivery city phenomenon change the urban experiences and lives of the research participants. In the case of shadowing, visual data, such as photographs and films, were collected alongside the researcher's observations, and an archiving procedure was carried out to document and arrange them. These archived data were used to examine, from a landscape standpoint, how themes and codes obtained from a prior phase operate in actual places (Figure 3).

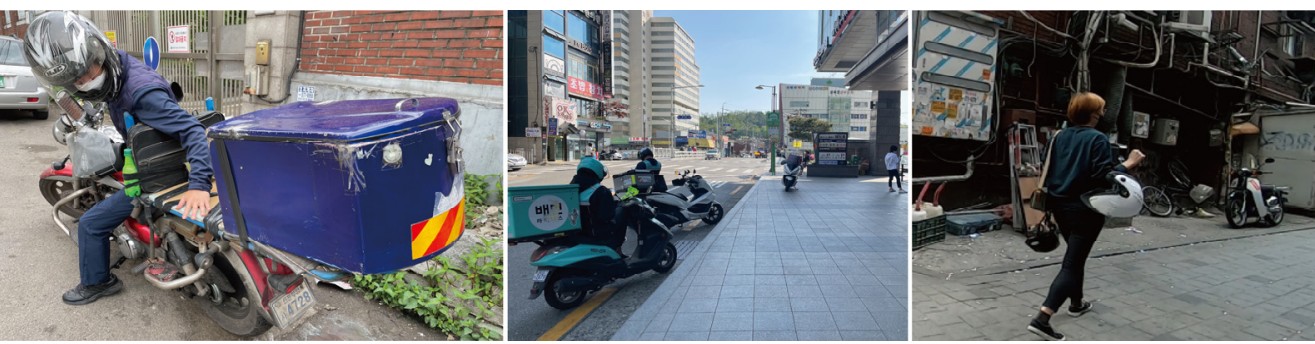

**Figure 3.** Field research record (source: created by the author).

Second, in order to comprehend the social perception and macro perspective of the phenomenon of the delivery city, articles containing the keywords "package and delivery" that were published by Korean media were examined. This research investigated 102,000 articles published between January 2019 and May 2022 by 19 domestic and economic daily newspaper companies to better understand the situation of delivery city. Text mining, direct citation of highly relevant articles, and relationship mapping between each word were conducted for analysis. Additionally, keywords frequently derived from news articles were also synthesised. The data derived from news articles were reflected and synthesized in the analytical stage of the qualitatively coded field-study data.

Using the codes and themes refined and derived from two phases, this research aimed to examine the delivery city phenomenon experienced by city actors as well as the social perception of this phenomenon. Using this as an indication to comprehend the existing situation, this study attempted to suggest tasks and implications in terms of urban planning reflecting the phenomenon of the delivery city.

## 4. Result

### 4.1. Derive Trends and Codes

As a trend in the media reports, the study's findings show that there is a connection between the delivery city and the COVID-19 outbreak. From there, the topic for classifying the full set of findings is obtained from keyword relation mapping. The interpretation of the field research data is synthesized and embodied with themes.

First, the media trends including the keywords "logistics" and "delivery" from May 2019 to May 2022 are shown in Figure 4. The number of articles including these keywords surged between February and March 2020, and then again in September 2020 and December 2020. This tendency seems to be consistent with the pattern of the COVID-19 pandemic in Korea. The first COVID-19 case was identified in Korea at the end of January 2020, and local infections began to appear from September 2020. Subsequently, between December 2020 and February 2021, the number of confirmed COVID-19 cases surged, resulting in a widespread national pandemic. The association between this media trend and the pandemic situation indicates that COVID-19 and the resulting pandemic made urban functions dependent on the mobility of things, thereby accelerating the delivery city phenomenon.

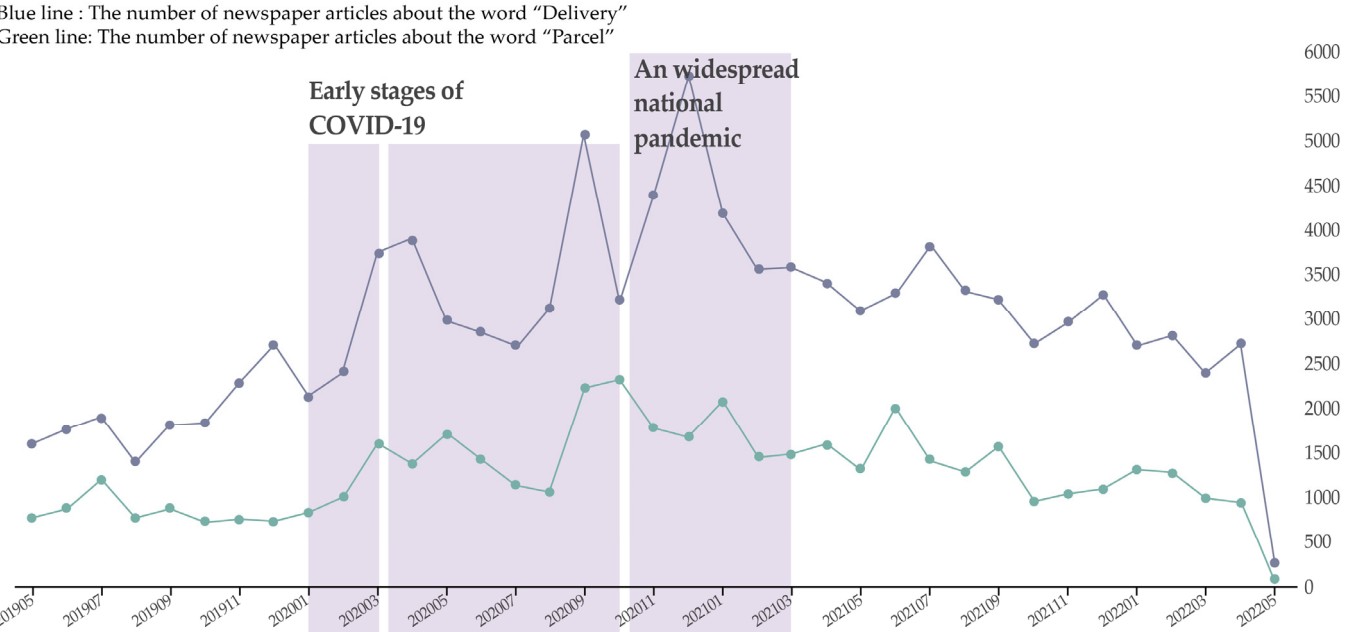

**Figure 4.** The change in the number of newspaper articles about "logistics" and "delivery" (source: created by the author with data from Korea Press Foundation BIG KINDS).

Figure 5 shows the keywords extracted from media articles. The upper left corner of the word map shows the most frequent keywords associated with the pandemic situation accelerating the delivery city phenomenon, such as COVID-19, confirmed cases, mobile, and non-face-to-face circumstances, and the "untact" situation. Additionally, the bottom left section shows the keywords pertaining to the working condition of deliverers, including online commerce platforms, delivery companies and unions, delivery and delivery companies, and labor law. The upper right corner illustrates the awareness of concerns associated with the phenomenon of the delivery city, such as safety, waste-related problems, and recycling. The bottom part includes keywords concerning major global logistics hubs, such as China, Busan, UK, and Japan, and places where objects and people interact, such as security guards, apartment complexes, and communities.

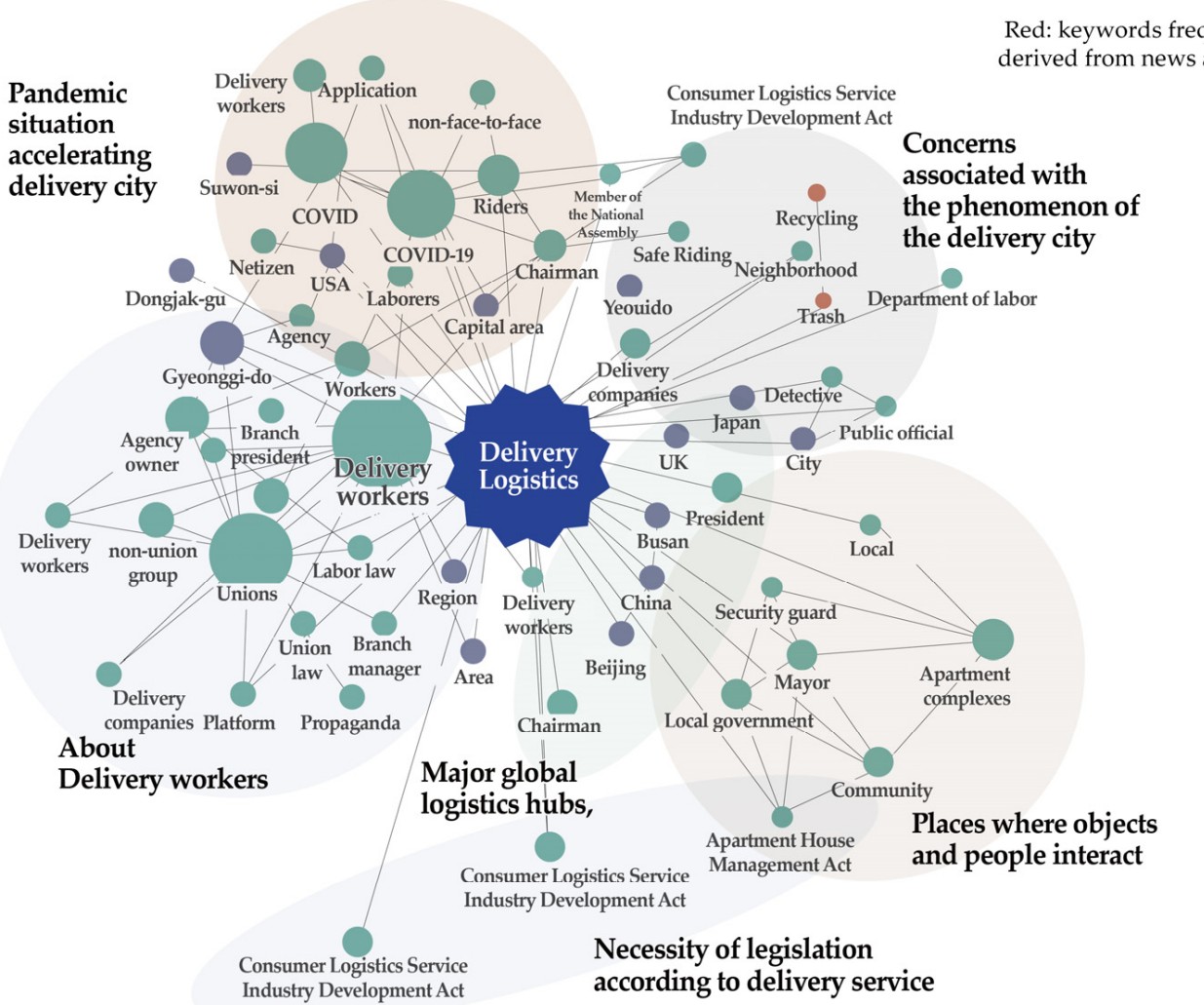

**Figure 5.** Keyword relationship mapping (source: created by the author with data from Korea Press Foundation BIG KINDS).

These keywords were synthesized in the interpretation phase for the coded interview data and research materials. As shown in Table 2, the synthesized research results were categorized into the following four themes

**Table 2.** Coded research results (source: created by the author).

| Theme | Code | Keyword |
|---|---|---|
| 1. Mobility and geographical awareness | 1.1 Spatial change | Door-to-door, gate, rider, fast quick service, road, accident |
| | 1.2 Imagined presence | Smart phone, platform, pandemic, my neighborhood, shopping mall, restaurant, SNS |
| 2. Boundary between private and public space | 2.1 Privatization of public space | Apartments, security guards, parcel crisis, the entrance, the underground parking lot |
| | 2.2 Invasion of private space | Crime, intrusion, women, safety, doors, conflict, my room, my house |
| 3. Spatial and social infrastructure of delivery city | 3.1 Labor space and roads | Trucks, cars, motorcycles, roads, subway toilets, a branch president, worker, agency |
| | 3.2 "Mobility of things" society | Platforms, accidents, traffic accidents, bikes, application, dangerous |
| 4. Sustainability and environmental issues | 4.1 Packaging materials | Garbage, market curly box, Styrofoam, tape, paper box, ice |
| | 4.2 The burden of disposing of trash | Boxes, many, recycling, plastic, recycling boxes |

*4.2. Mobility and Geographical Awareness*

A place can be defined as a space constructed based on relationships formed through connections such as repetitive sensations, attachments, and experiences of individual entities [30]. However, the mobility turn, which varies based on the high mobility of people and things, the fundamental unit of "placeness" and "spatial awareness" in the delivery city, is experienced repeatedly and instantaneously via mobility and fluidity as opposed to recurrent attachment and fixed coordinates. Consequently, in the era of mobility of things, spatial awareness and spatial experience are constructed along the movement of things rather than being based on a fixed space. This spatial awareness can be observed in the script of an interview with a delivery worker (C-3).

> *We are almost unable to rest during the delivery work. We never know when or where we may get a call requesting us to come to the restaurant. The whole street serves as both a place to rest and a place to work. A few kilometers in radius from where I am presently is all my area. (C-3)*

This illustrates that, in order to traverse space, the delivery person perceives each space as a referential, consecutive, and overlapped space, and that the delivery person does not form a based, fixed, or deep relationship with the space, but simply experiences the formation of a single movement.

This is consistent with the argument in [31], in that urban spaces in modern society are being changed into spaces simply for movement and passing, without any historical meaning or profound symbolism [32] to accommodate high mobility. Based on spatial fixity, the concept of place itself takes on the character of an event and transforms into a temporal and personal concept. In modern society, considering that communication occurs in a connected network and a continuously moving space rather than a fixed infrastructure in space, network thinking is stressed while the "physical division" of space loses its usefulness and value [18,31]. Therefore, the space that facilitates mobility serves primarily as a space for transit and movement and begins to seek continuous and rapid mobility. In this process, the "wall", the boundary that previously divided the space, becomes an impediment to rapid mobility [31].

This "placeness", which is altered by the city that allows such high mobility, is more pronounced in the delivery city. Urban spaces take on the characteristics of "a space of flows [14,31,33]" because they do not establish the physical boundaries in order to efficiently manage the mobility of things. As an example of the space of flows without

physical divisions, a worker (B-1) in a distribution center demonstrated how the distribution center operates to allow for rapid mobility.

> *Even when it's above 30 degrees outside, I have to wear padding to work. This place feels very separate from the rest of the world because the temperature is obviously different from the outside and it is located like an isolated island. It's huge on the inside, and thousands of people work there. (B-1)*

The interactions between the space and the actors, as well as the "placeness" that the actors perceive, vary as the structure of the spaces and actors in the delivery city change. First, awareness of physical space changes in accordance with changes in the physical mobility of each actor. As the primary actor, the delivery person, is outsourced, the movement, the mobility of other actors positioned at a fixed "point", reduces, and the vast urban space is perceived as a space of flows by the delivery person who executes the movement. Accordingly, the authors of [34] argue that "the street vanishes, and the place becomes virtual" as many social activities are performed via mobile phones. In this context, the "placeness" of a physical space, such as a street, is diminished in the delivery city phenomenon.

In contrast to the polarization of physical mobility, however, the network mobility of actors presents at each place increases through the "imagined presence". This is due to mobile networks and things reproducing spatial experiences via the actors' imagination. Lefevre illustrates the significance of residing in a home, a private space, in two ways. First, this refers to being immobile and fixed within the spatial category of a house. Second, it indicates that it functions as a "node point" in the process of constantly and intricately flowing energy into and out of the private space through "imaginable circuits". The function of such a node point proactively generates spatiality via the imagination of humans. In this sense, the fixed point and private space of the delivery city have network mobility along with the imagined presence. Therefore, the existing discourse stating that "placeness" is lost due to the development of mobility may be reinterpreted as "placeness" being rediscovered by recognizing the presence of virtual network space and things that link it to the actual world. Things that can be experienced in a certain space reconstruct that space through imagined presences, even if they are embodied in a completely different space. Things with a high level of network mobility evoke the recipient's imagination at the destination and generate "placeness" while moving from point to point. Using pizzas as an example, the manager of a fresh food platform company (A-1) demonstrated the process of generating "placeness" through an imagined presence.

> *The owner of Tombola \* bakes pizza in the factory on the second floor of the store, then quickly freezes it and sends it to us. Curly uses a full-cold-chain system that keeps food cold and frozen from the vehicle. Because of this technology, Tombola pizza tastes just like Seorae Village's one. Foods in the restaurant that you can see on social media are delivered quickly after being well packaged, right? I guess it is how we turn the house into the restaurant. (A-1)*

\* Name of a popular restaurant in Seoul.

The transformation of spatial functions in the delivery city diminishes the "placeness" and the significance of physical space. However, instead of diminished the significance of physical space, the actors of the delivery city, particularly consumers who become physically immobile as a result of the mobility of things, gain mobility via an imagined presence in the virtual network. Contrary to Putnam's argument that social capital declines as mobility increases, social capital actually transforms into network capital and new forms of trust and relationships are established. Similarly, virtual space is newly regarded as a significant "place" for actors and has significance, instead of physical "placeness" being weakened by the mobility of things.

This indicates that "placeness" in the delivery city, which has high mobility and highly developed information and communication technologies, should also be grasped in a relational manner throughout the planning phase. In other words, it is vital to analyze the

change in this relationship since "placeness" continues to change based on the relationship between actors, things, mobility movements, and actors.

### 4.3. Boundary between Private and Public Space

Paradoxically, the delivery city enhances the purpose and significance of private space. The work in [35] describes residing in a home as being immovable and fixed, or as energy moving in and out of the house through all circuits. In this perspective, the living space in the phenomenon of the delivery city serves as a fixed indicator and conduit for the internal or external flow of energy and things. Consequently, the gate, which is the entry to the conduit, plays a significant role as the dividing boundary between the private and public areas.

As a result of the presence of the deliverer who traverses between the inside and outside, the boundary between spaces becomes dynamic and fluid, generating movement and flow. As the objective of mobility is to alter the position of things rather than the position of people, the urban space of mobility, which is separated by a massive roof, develops the "wall", or the door, of private space in a significant manner. The front door, i.e., the "wall" in front of the stranger, opens for a purpose, and simultaneously closes with tense and quick communication in the air. In the era of mobility of things, the door, a private space that has not been observed, becomes socially and urbanely apparent and the most significant physical boundary.

> *So I try not to open the door. I didn't want to let people into the house where a woman lives alone If I open a little the door, someone can force their way in. (D-1)*

> *All the buildings have a common entrance gate. Normally, we should use a card key or enter the password. That means, we have to cross more than two gates. (C-2)*

> *When I wake up or when I come home, there are many delivery boxes in the front door. (D-2)*

Along with the mobility of things, the private space of the receiver inside the gate itself strengthens its significance. In an intimate private space, the consumers become physically immobile, yet, in terms of the network, they are constantly linked to various imagined spaces. In a hyper-connected society, given that the space where "phono sapiens" is most actively engaged in consumption activities is the bed, behaviors taking place in the everyday space, which was formerly a space for rest, reproduction, and family, are being diversified. The delivered objects enrich this experience in a private space. An office worker (E-1) and a university student (D-8) describe the process of altering the perception of private space based on the imagined presence and representation of experience provided by things.

> *If I don't use a smartphone, I can't eat, use banking services, listen to music, or even talk with friends. It's the basics of everyday life. I can't imagine life without (a smartphone) Most of my shopping is done while I'm in bed. Before I go to bed, I do it while lying down and looking at my phone. (E-1)*

> *It is written that if you deliver Starbucks, the place where you are will soon become a Starbucks, right? I think I don't have to go somewhere dangerous to drink (in the COVID-19 situation). My room is the safest café. (D-8)*

Particularly with the emergence of the non-face-to-face society, living spaces have come to serve work-related functions such as telecommuting in addition to the traditional housing and reproduction functions. Accordingly, as the hybrid feature of living space strengthens, the question as to how to determine the boundary between private and public areas is emerging as a crucial challenge for the delivery city.

First, the closure of a gate is generally understood in the context of gated communities, which are quickly gaining prominence in Korea. In the era of mobility of things, the gated community that is related to the apartment's gate enforces internal rules on other actors, thereby restricting the entrance of packages and deliveries. It has a characteristic of closure similar to "the ideal island for protecting private property" [36]. The stature of the

wall referred to as "the gate" hinders the movement of objects, such as by obstructing or circumventing the delivery person. In [36], Professor Park Insuk argued that this conflict is the result of the construction of a spatial structure that reinforces such closure, such as designating the ground as a pedestrian section, the presence of security guards, and installing locking systems [36]. The purpose of a closed spatial structure is to normalize the behavior of the inhabitants and the culture inside the door in a closed manner.

In Gwacheon, Ansan, and Suwon, an experiment is planned regarding the design of apartments in the manner of Paris that will allow the gated community and the city as a whole to cohabit [36]. Public spaces such as shopping malls, landscaping spaces, and community facilities that operated only within the gated community will be provided at the city level, and individual apartments will be built [37]. Although these options are structural alternatives that circumvent closure by altering the spatial structure, the experiment is still in its infancy and cannot address the closure of an existing residential complex. Consequently, solutions such as multi-structured delivery, including the last mile [38]), and the proposal to utilize unmanned delivery boxes are necessary, while adhering to the spatial norms of the existing gated communities and complementing the mobility of things. In addition, cultural improvements by which to resolve the norms and closures of gated communities could be a realistic alternative.

Furthermore, the space inside the door is vulnerable to safety concerns since it is accessible to another person—the delivery person. Since a license to access another person's space is an enticing circumstance for unlawful invaders, impersonation crimes are increasing. Because the delivered product serves as an access license for the delivery person, the recipient may be injured by criminals impersonating delivery people in cases when they open the door without caution [39].Regarding this situation of conflict and tension, an interview participant (D-5) describes the moment when he receives the products.

> *So, after that happened, I try to receive it in a non-face-to-face manner, and I turn on the TV loudly and wait 5–10 min before I open the door. Of course, it's cumbersome, but safety is the most important thing. (D-5)*

As the majority of interviewees in single-person female households in this study indicated, threats that occur at the boundary of these private spaces need to be considered more critically in the context of living space in the delivery city.

To prevent such crimes and to enhance security, the Korean government recommended a new spatial planning component for the construction of pedestrian facilities in residential areas. Spatial design methodologies such as CPTED (crime prevention through environmental design) or incorporating the process of distribution and delivery into specific guidelines might be considered as a crucial design alternative. Specifically, a design that aids in avoiding face-to-face contact by removing the vulnerable process of the door being opened may be an example. In addition, in a region with numerous single-person households and little space managers, safety delivery boxes can be installed a local police center with CCTV and emergency bells [40]. Furthermore, a "safe house set" consisting of a door auxiliary key, door opening center, window lock, and crime prevention window may be offered for the protection of women who live alone [41,42]. These solutions represent the reaction of urban design against daily threats in the delivery city.

### 4.4. Spatial and Social Infrastructure of Delivery City

The third characteristic of the delivery city is that it is functionally reliant on the mobility of things, and the public infrastructure is developing to facilitate the mobility of things. The characteristics of the spatial experience encountered in this space of mobility overlap several fields or areas of action. Specifically, the experience of public space, which works as a conduit of mobility and an intermediary point between the bases of each place, becomes highly overlapping and multidimensional. The public space of the delivery city is characterized by liminal space in which one area has an overlapping status with another. Consequently, conflicts and contentions between tasks occurring in the public space emerge, and the public space's fundamental function is jeopardized.

Bicycles occupying sidewalks and crosswalks being converted into waiting areas are representative instances of public spaces transformed into liminal space. This overlapping role of public space increases the efficiency of the delivery city, but also generates a conflict between the citizen and the delivery person who previously enjoyed the function of the public space, resulting in a decreased quality of life in the city as well as impingement on the walking rights of citizens. In addition, public space that serves as the delivery person's workplace is unstable, resulting in there being "nowhere to go even if they want to relax for just five minutes [43]".

> *We usually eat and sleep in the truck. There are no other options, even if it's dangerous. Because here is the only place. (C-2)*

> *In fact, we can search for toilets in the app. But the majority of open toilets are locked or unusable. During delivery, it takes a long time to reach a subway station toilet. So, riders make a joke that the ability to holding in urine is also an essential talent for a skilled rider. (C-5)*

In this context, when planning public spaces, it is important to consider the working environment of delivery persons, who are the central actors of the delivery city. The Seoul Metropolitan Government stated in December 2020, through the Second Basic Labor Policy Basic Plan, that it intends to establish shelters for delivery workers in all districts by 2023. As of April 2021, there are five shelters including Hapjeong Shelter (opened in November 2017), Seocho Shelter (opened in March 2016), Bukchang Shelter (opened in June 2017), Sangam Shelter (May 2018), and Nokbeon Shelter (July 2019), which are open until 6 p.m. and do not operate on weekends [43,44]. The utility and number of facilities are inadequate in light of the working environment of delivery workers who work regardless of the time of day, including weekdays and weekends, and the growing number of delivery persons. However, the authorities say that it is difficult to operate the existing facilities due to the lack of a related budget. Goyang City plans to open a container-style shelter with a 24 h unmanned security system on 14 December 2021, recognizing that it is difficult to administer and manage a delivery shelter that leases a facility and employs personnel [45] In addition, it is necessary to analyze the transportation facilities that regulate the parking and stopping of delivery drivers. There is a requirement that loading zones do not interfere with the main road owing to the parking and stopping of trucks and two-wheeled vehicles. Accordingly, the work in [46] categorizes parking and stopping facilities as street parking lots, bus stop facilities, taxi stands, stops for general vehicles, and operation parking spaces for two-wheeled vehicles in response to the varying demands of urban traffic. It is recommended that these parking spaces be constructed in a type of a bay, without diminishing walkways, so as not to impede primary traffic [46,47].

Additionally, the institutional and cultural foundation for the mobility of things should be improved. As delivery expands in the "untact" era, more than 20,000 two-wheeled vehicle accidents, such as those due to unsafe driving and signal violations, occur annually [48]. This is due to the structure of the businesses in which employees of the mobility of things are outsourced for each movement and deliverers are required to deliver more things more quickly, rather than reducing customers' time and space restrictions. Deliverers C-1 and C-8 show how the structure of a business creates a culture in which the infringement of traffic regulations is approved implicitly.

> *We even make a joke that the red light means a red signal to go fast, and the blue light is a signal to go safely. It's been 30 years since I've worked it, but my friend also passed ways. (C-1)*

> *Delivery drivers also want to drive safely. When it rains, the road is especially slippery. I feel that I'm really going to die. However, if the required time is exceeded, customers give a low rating, and the app limits the requests. We have to run fast to work and make money. People who order should also know that it's inevitable that we break the law. (C-8)*

Consequently, noncompliance with safety requirements accounted for 53.6% [48] of motorcycle accidents, resulting in a dangerous trade-off between safety and speed. However, platform companies lack their own training programs, substitute face-to-face

safety teaching with videos, and exam answers are shared through the Internet, resulting in the training programs having little use [49].

In response, the Seoul Metropolitan Government and the Korea Transportation Safety Authority provide free two-week safety experience education for delivery riders from November 2021 [50]. In addition, the Korea Delivery Riders Association and the Korea Platform Freelance Labor Mutual Aid Association regularly educate delivery riders on traffic safety [51], but the participation rate is still low since it is not mandatory. With the assistance of businesses and the private sector, it is necessary to build a culture of safe delivery. In this regard, the authors of [52] argue that there is "a structural problem that cannot rely only on individual compliance" regarding the reduction in vehicle accidents of riders, and "a business model in which risky driving makes more money" should be changed into "a business model in which safe driving makes more money." He also emphasizes the need for the intervention of businesses, the private sector, and the government in this process.

In conclusion, to avoid public spaces becoming transformed into liminal spaces, it is vital to examine the purpose and occupancy of public and private spaces in the city as a task of physical space design. First, from the capitalist perspective, it is important to rearrange public places that interrupt the walking rhythm in accordance with the "de-infrastructure phenomenon" that continues to redirect existing infrastructure. As the demand and supply of the mobility of things continue to increase along with the economies of scale, cities that lack infrastructure such as parking places for the mobility of things and traffic rules are likely to be unable to manage these. Consequently, it is vital to introduce resting spaces for delivery workers, parking and stopping zones, and the norms of the delivery city in order for it to operate as a public place that can preserve the rhythm of each city.

As the public space becomes a liminal space, space limits are reduced and the boundaries between private and public space are demolished, providing an opportunity to effectively manage the limited resources and functions of the city. Private and public spaces whose boundaries are fluid with the mobility of things may be designed as multidimensional and flexible vast spaces that facilitate diverse movement patterns and mobile habitation. Commercial space, which is functionally weakened owing to the mobility of things, may operate as a place of integration and diversity by establishing interaction between the façade and public functions via flexible city management.

*4.5. Sustainability and Environmental Issues*

Objects in the delivery city phenomenon invariably require packaging material for movement; hence, the large amount of packaging material produces environmental difficulties. Groceries, in particular, which are quickly increasing as a result of COVID-19, necessitate more packing materials in order to maintain freshness and ensure safe delivery. This is due to the fact that foodstuffs are viewed as "things" in the process of the mobility of things and receive both large and minor shocks from the distribution center to customer delivery; they also require being maintained at the proper temperature and should contain packaging that is easy for the mobility of things workers to carry. As a result, the delivery city has a negative reputation as "a civilization overflowing with rubbish as delivery apps develop" [53]. These are issues shared by interviewee D-1, who appreciates utilizing a mobility of things platform.

> *I was hesitant to employ (a fresh food delivery service) because ice packs were normally discarded as general garbage after usage. Not to mention the Styrofoam containers. First and foremost, the size is far too large. Aside from environmental concerns, I dislike a lot of waste. (D-1)*

To regulate temperatures and prevent damage, most fresh food deliveries use expanded polystyrene–Styrofoam boxes. These have great insulating and buffering qualities and are commonly used for the transportation of food products since germs cannot grow in them. However, they do not naturally degrade, thus putting a strain on the environment. A situation in which packing materials are filled in recycling boxes is not only inconvenient to manage, but also places a psychological burden on environmentally conscious consumers.

Indeed, according to [54] containing a report on throwaway products owing to delivery and delivery, "2021 Plastic Stay-home Survey: The Face of Disposables", as the dangers to society spread and the "untact" lifestyle develops, delivery and food delivery have surged, and plastic and vinyl waste have grown by 14.6 and 11%, respectively. Furthermore, food packaging materials accounted for 78.1% of household plastic waste [54], accounting for 60,331 of 77,288 plastic wastes in 841 studied households. In Korea last year, around 3.3 billion parcels were generated [55], with 7.14 million pieces of delivery garbage generated per day [56]. Korea has the world's third highest per capita plastic garbage emission of 88 kg per year [57] (Maeil Economic Daily, 6 December 2021), and disposable containers increase carbon emissions by 35 times [58]. This is far too high to be disregarded as an unavoidable by-product of the delivery city and the era of mobility of things, and it is truly "labor to toss away" [59]. Interviewees D-5 and D-6 also expressed concern about the burden of disposing of waste from the mobility of things.

> *I ordered five small items, and five boxes arrived. Every week, the garbage can becomes full. If you simply walk out and buy it, you will be given an envelope . . . It's quite an effort to get rid of it. (D-5)*

> *I feel awful for the world every time I open a package and look at the garbage bin, right after I receive food or parcels . . . Now, I just go out and buy them or attempt to cook if I can. (D-6)*

To address this waste issue, organizations in the manufacturing and distribution stages must first implement technological changes in packaging materials. Eco-friendly plastics, paper, and oxidized biodegradable films that decompose in nature or packaging design that minimizes the use of subsidiary materials such as tapes used to seal products [60] macroscopically reduce environmental problems in cities that rely on the mobility of things.

It is necessary to improve the culture at the receiving and processing stages, such as using previously used packaging materials or self-packaging using home packaging containers. To this end, a service that collects and cleans multi-use containers during the delivery process [61], a refill station where you can buy simply the contents without packaging [62], and a marketing strategy reimbursing clients who self-package is required.

The environmental problem produced by the rapid increase in the mobility of things is a key urban problem that must be addressed in tandem with the advancement of logistics technology, and it is a factor that threatens the sustainability of the delivery city. Therefore, discussion at the policy establishment stage is critical because it is hard to fundamentally tackle this problem solely through consumer and business initiative. In fact, in order to reduce the amount of waste that has increased since COVID-19, the government has been partially amending the provisions, with initiatives such as "Expanding the target of single-use product regulations" and "Displaying products using recycled materials and the compulsory purchase of public institutions", which are described in "Enforcement Decree of the Act on the Promotion of Saving and Recycling of Resources" [63].

Furthermore, legal underpinnings for the continuation of the delivery city are being created, including multi-use packaging materials and regulations for excessive parcel wrapping [64]. Additionally, short-term, mid-term, and long-term policies for environmental challenges in the delivery city are being addressed, such as setting a target of deliveries being entirely carried out using electric delivery vehicles in Seoul by 2025 [65]). These discussions are ultimately tied to the delivery city's sustainability; thus, a more active interest and reflective attitude toward logistics, which have become natural in the era of the mobility of things, are necessary.

## 5. Conclusions

Through a detailed review of media articles and qualitative research in the field, this study investigated the "untact" society triggered by COVID-19 constraining human mobility and envisioned the delivery city phenomenon from both the actor and the social points of view. First, the identification of trends in media articles shows that social interest in

delivery and mobility of things increases with the emergence of COVID-19. In the keyword relationship mapping of media articles, it was possible to grasp the overall social perception of delivery service. Specifically, it can be divided into the following: (1) interest in untact delivery services due to the pandemic; (2) labor environment problems for delivery workers due to soaring delivery volume; (3) consciousness of sustainability regarding by-products, such as safety, waste, and recycling; and (4) awareness of urban spaces that receive parcels, such as at an apartment or on the front door.

As a result, four urban tasks were derived through comprehensive coding. First, based on the delivery city concept, cities quickly became highly mobile, and the function of urban space was transformed into a space of flow. Accordingly, the meaning of physical space weakened, captured in the decline in the placeability of space, for example. At the same time, the interaction between the space and actor also changed due to the imagined existence provided by objects.

This has led to the notion that urban planning should precede the understanding of the spatial experience of actors who change in the future. More specifically, it is necessary to prepare a spatial alternative to conflict interaction occurring at the boundary between private space and public space, which is a conduit and gateway for objects to cross. There is also a need to establish spatial and social infrastructure that becomes urgent when a delivery city becomes an important phenomenon in urban society. Furthermore, to create sustainable delivery cities, their by-products, such as garbage and environmental problems, must be addressed, and tasks such as deriving policy and legal alternatives and urging urban residents to improve their awareness are suggested.

Along with tasks, the most important city challenge facing the delivery city phenomenon is to recognize that the function, appearance, and actor interaction of the city are changing according to the delivery service. It means that delivery service is not a simple movement of objects, but a significant factor that changes the meaning of urban space. The key to making the delivery city sustainable will be this expanding perception, which leads to the above-mentioned policy alternative and public awareness about delivery.

This study has some limitations. The restriction of the spatial scope to the metropolitan areas of Korea and the spatial practice of actors were not dealt with in detail. Mandl [66] proposed an alternative to platform labor constituting modern delivery cities by focusing on the specific situation of workers, and Lizzie [7] also proposed in-depth research and questions about platforms and delivery workers, which are heterogeneous systems. Hence, it is necessary to differentiate the actors of the delivery city in detail to grasp their spatial practice and experience. In addition, as delivery cities are mainly formed through platforms, called apps, research on an entire platformized city, such as those of Graham [6] and Rose et al. [67], needs to be expanded further with the delivery city approach.

The significance of this study, however, can be regarded as the fact that it studied the city in the context of the mobility of things, which is becoming increasingly important around the world; additionally, this study used an active research process regarding sustainability and proposes that developing the phenomenon of the delivery city can progress in a more positive direction by proposing urban planning implications based on the phenomenon experienced in modern cities. Furthermore, the perspective of interpreting space with a focus on the mobility of things, as opposed to the traditional human-centric approach to perceiving spatial geography, will have significant ramifications in terms of broadening the perspective of interpreting space and cities.

During the pandemic, the entire globe recognized that the mobility of things is an essential service in modern cities and is in line with the flow of time. Furthermore, the delivery city phenomenon has been identified as a significant phenomenon that is transforming the modern urban form as a whole. However, urban and spatial responses are falling behind the continuously changing reality. This study proposes that future research on the mobility of things and the phenomenon of the delivery city should be openly debated within the academic community along with a perspective that takes the delivery city's urban design and social concerns into account.

**Funding:** This research received no external funding.

**Institutional Review Board Statement:** Not applicable.

**Informed Consent Statement:** Not applicable.

**Data Availability Statement:** Not applicable.

**Acknowledgments:** I acknowledge that this article was written based on the author's thesis at the Graduate School of Environmental Studies Seoul National University. I would like to express my thanks to my thesis reviewers for helping me develop the research.

**Conflicts of Interest:** The author declares no conflict of interest.

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
