# Peer review of "Challenges Facing the Delivery City Phenomenon after the COVID-19 Pandemic"

_sustainability, doi:10.3390/su14159243_

Round 1

Reviewer 1 Report

1-     The subject is new for me, however in the abstract I could not understand the research problem, the novelty, and the method by which you want to solve the problem.

2-     I suggest using a qualitative method like Grounded Theory in your methodology.

3-     Use more recent papers in the literature review.

4-     The results section needs to be revised. Currently, the format is not similar to a scientific paper.

Reviewer 2 Report

Reviewer Comments:

General comments:

It is my pleasure to review the paper entitled “Challenges Facing the Delivery City Phenomenon after the COVID-19 Pandemic: An Empirical Study of the Mobility of Things in the Republic of Korea ”. The manuscript has multiple issues and could not recommend it for publication and need extensive revision. 

I present the following comments that can help to improve the paper:

Detailed comments: 

Title: 

Authors need to revise the title.

 Abstract

The authors should write the main problem and then start how this study is useful to tackle those issues. 

Introduction

The introduction section needs thorough revision,

  1. What are your research questions and hypotheses? What are the important findings expected by the readers? What are research gaps in the past, and what are your contributions to improve them? 
  2. Some of the recent and important published literature is missing, please add recently published articles from 2019-2021 and extend the introduction section. 

Methodology: 

  1. In the Mathematical Modeling, please elaborate on this section so that an independent researcher could adopt this research. 
  2. Figure 1 is not clear. 
  3. No detailed description of method, please elaborate this section. 

Result and discussion: 

  1. How the keyword relationship mapping was analyzed here using model, could you please elaborate on it?
  2. I would suggest the authors to explained the result in more detail,in the current form couldnot understand what is the main purpose of this articles and which result are important from this research. 
  3. In addition, the authors could extend the discussion section by comparing their results with previously published studies.
  4. The authors can also recommend some future research work.
  5.  

Conclusion

  1. What are the limitations of this study?

Best,

Reviewer 3 Report

Dear Author,

the proposed topic. in particular the impact of COVID19 pandemic into the mobility,  is undoubtely interesting.

But, at the same time, I feel the need to highlight how some improvements are required. In particular, even if  the research work is well explained in terms of micro and macro data collection, the methodology section should  explain in detail the tools you have used to achieve the target informations. An annex at the end of the paper should be interesting to show to the readers the questions submitted to the selected respondents.

About the purpose of the paper, the link between the information acquired and the possibility for it to be used as a tool for urban planning, is not shown.

In conclusion, I think that once clarified the above mentioned aspects, the work can be accepted.

Warmest regards

Round 2

Reviewer 1 Report

No new comment!

Author Response

Point 1: No new comment!   Response 1: I would like to thank you for your review and comment.  

Reviewer 2 Report

I present the following comment:

Which one author thick is the most important city challenges facing the delivery city phenomenon and I would suggest authors to provide some of key points to make is a sustaintable. 

Best, 
